# Role of PEG35, Mitochondrial ALDH2, and Glutathione in Cold Fatty Liver Graft Preservation: An IGL-2 Approach

**DOI:** 10.3390/ijms22105332

**Published:** 2021-05-19

**Authors:** Raquel G. Bardallo, Rui Teixeira da Silva, Teresa Carbonell, Emma Folch-Puy, Carlos Palmeira, Joan Roselló-Catafau, Jacques Pirenne, René Adam, Arnau Panisello-Roselló

**Affiliations:** 1Experimental Pathology Department, Institute of Biomedical Research of Barcelona (IIBB), CSIC-IDIBAPS, 08036 Barcelona, Spain; rgomezbardallo@ub.edu (R.G.B.); rui.teixeira@iibb.csic.es (R.T.d.S.); emma.folch@iibb.csic.es (E.F.-P.); arnau.panisello@iibb.csic.es (A.P.-R.); 2Department of Physiology, Faculty of Biology, Universitat de Barcelona, 08028 Barcelona, Spain; tcarbonell@ub.edu; 3Center for Neuroscience and Cell Biology, University of Coimbra, 3004-517 Coimbra, Portugal; palmeira@ci.uc.pt; 4Department of Life Sciences, University of Coimbra, 3004-517 Coimbra, Portugal; 5Department of Abdominal Transplant Surgery, University Hospitals Leuven, 3000 Leuven, Belgium; Jacques.pirenne@uzleuven.be; 6AP-HP Hôpital Paul Brousse, UR, Chronothérapie, Cancers et Transplantation, Université Paris-Saclay, Villejuif, 91190 Paris, France; rene.adam@aphp.fr

**Keywords:** fatty liver, PEG35, IGL-1 solution, ALDH2, 4-HNE, nitric oxide

## Abstract

The total damage inflicted on the liver before transplantation is associated with several surgical manipulations, such as organ recovery, washout of the graft, cold conservation in organ preservation solutions (UW, Celsior, HTK, IGL-1), and rinsing of the organ before implantation. Polyethylene glycol 35 (PEG35) is the oncotic agent present in the IGL-1 solution, which is an alternative to UW and Celsior solutions in liver clinical transplantation. In a model of cold preservation in rats (4 °C; 24 h), we evaluated the effects induced by PEG35 on detoxifying enzymes and nitric oxide, comparing IGL-1 to IGL-0 (which is the same as IGL-1 without PEG). The benefits were also assessed in a new IGL-2 solution characterized by increased concentrations of PEG35 (from 1 g/L to 5 g/L) and glutathione (from 3 mmol/L to 9 mmol/L) compared to IGL-1. We demonstrated that PEG35 promoted the mitochondrial enzyme ALDH2, and in combination with glutathione, prevented the formation of toxic aldehyde adducts (measured as 4-hydroxynonenal) and oxidized proteins (AOPP). In addition, PEG35 promoted the vasodilator factor nitric oxide, which may improve the microcirculatory disturbances in steatotic grafts during preservation and revascularization. All of these results lead to a reduction in damage inflicted on the fatty liver graft during the cold storage preservation. In this communication, we report on the benefits of IGL-2 in hypothermic static preservation, which has already been proved to confer benefits in hypothermic oxygenated dynamic preservation. Hence, the data reported here reinforce the fact that IGL-2 is a suitable alternative to be used as a unique solution/perfusate when hypothermic static and preservation strategies are used, either separately or combined, easing the logistics and avoiding the mixture of different solutions/perfusates, especially when fatty liver grafts are used. Further research regarding new therapeutic and pharmacological insights is needed to explore the underlying mitochondrial mechanisms exerted by PEG35 in static and dynamic graft preservation strategies for clinical liver transplantation purposes.

## 1. Introduction

Unhealthy lifestyles associated with alcohol consumption and inappropriate diet, along with other factors such as aging, are responsible for the accumulation of fat in the liver, which leads to varying degrees of undesirable hepatic steatosis [1]. Considering the urgent lack of organs for transplantation, physicians have been obliged to take advantage of fatty livers [1,2] to increase the donor pool and thus shorten waiting lists for clinical transplantation [3]. However, it is clear that steatotic livers show higher vulnerability against cold ischemia and reperfusion injury [4], and their use increases primary failure and compromises graft outcome after transplant [4,5].

Ischemic injury in liver transplantation is associated with mechanical organ manipulation by the physician prior to transplantation. The process includes organ recovery, washing of the graft, cold storage in an organ preservation solution, and finally, rinsing of the graft. Due to the combined action of ischemia and cold preservation, the graft may undergo damage. As a result, the cumulative injury in both mandatory steps before transplantation must be minimized in order to achieve the recovery of the graft’s function after liver transplantation, especially in the case of steatotic grafts [1,2,3,4].

The most frequently used preservation solutions for liver transplant are UW, HTK, Celsior and, more recently, IGL-1 [6,7]. Certain limitations regarding the use of HTK [8,9] have been pointed out by the United Network for Organ Sharing (UNOS) and the European Liver Transplant Registry (ELTR). IGL-1 emerged as a good alternative to UW solution, which is the gold standard [9]. The only differences in their composition are the oncotic agent (HES for UW and PEG35 for IGL-1) and the reversal of Na+/K+ concentrations conferring IGL-1 the property of extracellular low potassium solution [10,11]. In addition, replacing the HES present in UW for PEG35 lessens red blood cell aggregation and favors organ rinsing, preservation, and perfusion [12]. All these changes in IGL-1 have shown to provide benefits in clinical liver transplantation in terms of reducing early allograft dysfunction [13] and improving graft survival according to the European Liver Transplant Registry (ELTR) [9].

Moreover, PEG35 presence in rinse solution for graft washing out promotes several cytoprotective factors, conferring hepatoprotection [14]. This includes the generation of nitric oxide (NO) whose vasodilation properties counterbalance the microcirculation disturbances in fatty liver grafts, favoring graft preservation and revascularization [15,16,17,18]. Keeping in mind the beneficial PEG35 properties, we recently proposed the use of IGL-2 solution (containing PEG35) as a suitable perfusate for dynamic hypothermic oxygenated strategies (HOPE) with promising results [19,20]. This might improve the only perfusate for machine perfusion currently available, Belzer-MPS and generics, containing HES [21], given that the use of a unique solution, such as IGL-2, for static preservation and machine perfusion (MP) would facilitate logistics and avoid the mixture of different solutions [19,20,21] when both techniques are combined. This is the criterion by which we evaluated the IGL-2 benefits in static hypothermic preservation in the present study.

Aldehyde dehydrogenase-2 (ALDH2), a liver mitochondrial enzyme that was initially implicated in the liver alcohol metabolism, has been associated with the pathophysiology of ischemia–reperfusion injury in several organs, including the liver [22,23,24]. Several authors reported that the use of Alda-1 (an activator of ALDH2) protects against liver ischemia–reperfusion injury (IRI) in the rat [25,26], but currently, no evidence exists on the direct PEG35 effect as a regulator of mitochondrial ALDH2 in cold old ischemic preservation strategies, although its cytoprotective action was indirectly evidenced when different organ preservation solutions were used for cold storage of fatty liver grafts [23,24]. ALDH2 could play a constitutive housekeeping role essential for the development and regulation of recycling and survival processes as those occurring in cold ischemia preservation [22,27].

With this in mind, we explored the relevance of PEG35 in IGL-2 solution (Table 1) on ALDH2. In this communication, we demonstrated for the first time how the direct effects of PEG35 on mitochondrial ALDH2 contribute towards maintaining mitochondrial functionality during ischemia preservation. Mitochondrial ALDH2 could act as a gatekeeper of ROS overproduction protecting the liver graft from the damaging effects of transient aldehydes produced [22,23,24] besides other additives in preservation solutions such as labile glutathione [28] that play an important role against oxidative stress.

The results reported here reveal the superior antioxidant capacity of IGL-2 (due to ALDH2 combined with increased glutathione content) against the ischemic insult during graft preservation, presenting an interesting tool to be considered for improving hypothermic fatty liver preservation by using static and dynamic approaches.

## 2. Results

The presence of PEG35 and glutathione (group IGL-1) is determinant for preventing liver cold ischemic injury (AST/ALT) and mitochondrial damage (GLDH) during fatty liver graft preservation and the prevention of energy breakdown during cold storage, at 4 °C during 24 h. As revealed in Figure 1, a higher ATP content is shown in liver preserved in IGL-1 solution (containing PEG35 as oncotic agent) than in liver preserved in IGL-0 (the same solution as IGL-1 but without PEG35).

Since PEG35 in IGL-1 preserved liver mitochondrial status during cold ischemia insult during experimental models [23], we evaluated its effect on mitochondrial ALDH2 and compared it to IGL-0 (which is the same as IGL-1 but without PEG35 in its composition). As shown in Figure 2, PEG35 presence in IGL-1 promoted significant increases in ALDH2, contrasting with the significantly lower levels of ALDH2 found in fatty liver grafts preserved in IGL-0 (without PEG35). (Figure 2).

Considering the relevance of PEG35 concentration according to the previously known results, we expanded our study to one additional group using IGL-2 solution [19,20]. We focused mainly on parameters relevant to the mitochondria (such as ALDH2) that regulate other cytoprotective responses. As demonstrated in Table 1, the IGL-2 solution is mainly characterized by higher concentrations of PEG35 and glutathione when compared to IGL-1 (Table 1).

Firstly, we analyzed transaminases (AST/ALT) release and mitochondrial damage (GLDH), from which the total liver damage could be inferred. As shown in Figure 3, the PEG35 presence was a major factor in preventing transaminases and GLDH release with a dose-dependent PEG35 tendency. Although no significant differences between IGL-2 and IGL-1 were observed (except for AST), the presence of PEG35 seems to be a determinant factor in preventing the release of transaminases and GLDH in a dose-dependent manner.

With this in mind, we evaluated the incidence of PEG35 on the energy breakdown prevention during cold storage by measuring ATP content in preserved livers. Data reported in Figure 4 revealed higher ATP levels in PEG-containing solutions and evidenced the PEG35-dependent energy breakdown prevention during cold preservation.

Additionally, and considering the cytoprotective autophagy as a recycling mechanism of nutrients and energy to cope with stress situations existing in cold ischemia conditions [27], we evaluated Beclin-1 and LC3B as autophagy marker [29]. Beclin-1 and LC3B levels correlated with the tissue ATP levels in preserved livers in IGL-0, IGL-1, and IGL-2. As shown in Figure 4, there was a significant upregulation of cytoprotective autophagy in the IGL-2 group compared with the others, which correlated with ATP levels in PEG35 groups, showing a positive tendance in ATP prevention for IGL solutions.

We correlated mitochondrial aldehyde dehydrogenase 2 (ALDH2) with the toxic aldehydes (4-HNE) and oxidized protein (AOPP) levels. Figure 5 shows an increase in the expression of the ALDH2 enzyme, which is concomitant with a decrease in the 4-HNE protein adducts formation, notably decreasing the levels of oxidized proteins going from IGL-0 to IGL-1 and IGL-2. When comparing IGL-0 and IGL-1, which only differ in the presence or absence of PEG, it can be seen that ALDH2 is augmented, the 4-HNE protein adducted, and AOPP decreased solely by the effect of PEG.

Therefore, to observe the impact of glutathione on the reduction of oxidative stress (and not only by PEG35 itself, which happens when comparing IGL-0 and IGL-1), reduced glutathione was measured. There was a significant difference of reduced glutathione in IGL-2 solution, which was to be expected due to its initial increased dosage; however, high variance in the IGL-1 group might suggest that part of its glutathione (if compared to IGL-0) might be spared (therefore, not oxidized) due to other antioxidant capacities derived from PEG35 (Figure 6).

Finally, we evaluated the direct effect of PEG35 on NO production and oncotic pressure to avoid oedema. As shown in Figure 7, we evidenced respective augments in NOx products and endothelial NO synthase expression levels. Significant (endothelial NO synthase) eNOS activity and NOx levels were observed for PEG35 groups (IGL-1 and IGL-2). This fact was correlated with a significant upregulation of mitochondrial ALDH2 expression in these groups. A positive trend towards increased eNOS expression and NOx levels in IGL-2 vs. IGL-1 was observed, but statistical differences were not found.

## 3. Discussion

It is well known that fatty livers are more susceptible to cold ischemia during preservation than normal livers. Therefore, the aim of our research was to increase the performance of their preservation by developing a new solution (IGL-2).

PEG35 is the oncotic agent present in IGL-1 solution [11], and its protective role in IRI [30], rinse solution [14], and cold storage [23,24] was previously described. It is an established fact that any preservation solution must contain antioxidants to counteract oxidative stress during IR. Considering this, the IGL-2 benefits reported here for hypothermic static preservation are consistent with its suitability to be used in HOPE over the currently used perfusates (Belzer-MPS or its generics) [19,20]. Both reasons constitute a solid basis for simplifying the logistics and avoiding mixing different solutions when both static and HOPE need to be combined. This is especially interesting for rescuing fatty liver graft liver transplantation purposes [21].

PEG35 solutions (IGL-2 and IGL-1) are related to reduced AST/ALT and especially to GLDH, showing better mitochondrial protection, as previously described [14,19,20]. PEG35 improved mitochondrial machinery and ALDH2 functionality, reducing graft cold ischemic injury [23,24]. Consequently, this calls for an exploration of the underlying protective mechanisms by which PEG35 confers mitochondrial protection on fatty liver grafts preserved in IGL-1 and IGL-2 solutions whose mechanisms substantially differ from the ones induced in HOPE [31], where the transient oxygenation during hypothermic preservation is responsible for maintaining mitochondrial activity function at basal levels. This confirms the relevance of organ preservation strategies on the mitochondrial status and their subsequent benefits in liver transplantation, as recently published by Horvath et al. [32].

Recently, it has been reported that aldehyde dehydrogenase 2 ALDH2 activation is linked to protective mechanisms in several organs, such as the heart, brain, kidney, and intestine [22,33]. The cardioprotective and neuroprotective role of ALDH2 in myocardial ischemia–reperfusion has been demonstrated [34], with recent evidence showing that ALDH2 inhibition alters endothelial functions along with a deterioration of bioenergetic functions [35]. ALDH2 arises as an important gatekeeper of ROS overproduction, making the cell more tolerant to it [36]. In fact, the main function of mitochondrial ALDH2 is to protect mitochondria and cells from the damaging effect of aldehydes (by oxidizing the substrates into their corresponding non-toxic carboxylic acids), which are involved in the oxidative stress associated with IRI [22]. Zhang et al. [25] demonstrated that Alda-1, an ALDH2 activator, protects the liver against warm IRI preventing oxidative stress. In this sense, our work revealed that the oncotic agent PEG35 could be considered as an enhancer of mitochondrial ALDH2 upregulation, whose underlying protective mechanisms against cold ischemic insult have not been assessed in depth [22].

The prevention of liver injury exerted by solutions containing PEG35 contrasts with the injury observed in livers preserved in IGL-0 (without PEG35) with depleted energetic levels after 24 h of cold storage. Remarkably, the comparison of IGL-1 to IGL-0, which only differ in the presence or absence of PEG, suggests that PEG35 by itself leads to an ALDH2 upregulation. This is consistent with the IGL-2 solution, where the augmented PEG35 content further prevented oxidative stress through ALHD2 upregulation and promoted cytoprotective autophagy.

Nevertheless, the total antioxidant effects of the IGL-2 solution are mediated by the contribution of ALDH2, besides the antioxidant action of glutathione present as an additive, which should be considered as well. Thus, the higher IGL-2 antioxidant capacity compared to that of IGL-1 (due to increased PEG35 and glutathione concentrations) is also reflected by preventing AOPP and 4-HNE protein adduct formation in PEG35 solutions (IGL-1 vs. IGL-0) and, even more importantly, with the presence of increased PEG35 and glutathione content (IGL-2 vs. IGL-1). Indeed, IGl-1 shows more ALDH2 than IGL-0, and more IGL-2 than IGL-1, suggesting that ALDH2 increases with PEG35 in a concentration-dependent manner.

In addition, it must be considered that the increased expression of eNOS synthase increased the hepatoprotection mechanisms during cold preservation [15]. The beneficial effects of PEG35 on ALDH2 mitochondrial machinery are also increased by the concomitant presence of glutathione to prevent the action of toxic aldehyde adducts (4-HNE) and lipoperoxide generation [22] associated with hypothermic storage. Alternatively, PEG35, through the increased e-NOS activation and subsequent NO generation, prevents the microcirculatory disturbances that occur in fatty liver graft revascularization [37]. Furthermore, some of this NO could act as a scavenger of ROS, thus reducing the number of oxidizing particles. IGL-2 is a suitable alternative solution for increasing cold graft preservation strategies when static cold preservation and HOPE need to be combined.

In conclusion, we demonstrated the relevance of the oncotic agent PEG35 in modulating the redox state through mitochondrial ALDH2, thus reinforcing the protection mechanisms of fatty liver graft in cold preservation in combination with glutathione. This could improve the preservation of fatty liver grafts and may help to design new static and dynamic preservation strategies using PEG-containing solutions/perfusates.

Further, in-depth research should be conducted to clarify the role of mitochondrial ALDH2 and its direct relationship with polyethylene glycols features as efficient tools for preventing IRI.

## 4. Materials and Methods

### 4.1. Animals

Homozygous (obese (ob)) Zücker male rats aged 16–18 weeks were purchased from Charles River (Charles River, Lyon, France). They were housed in a temperature-controlled environment (25 °C) with a 12 h light/dark cycle and provided water and standard chow ad libitum. The rats presented a rate of steatosis between 60% and 70%. All procedures were carried out according to the EU rules for animal experiments (EC guideline 86/609/CEE) and were approved by the University of Barcelona’s Ethics Committees for Animal Experimentation (#483/16). The animals underwent general anesthesia with isoflurane inhalation.

### 4.2. Experimental Groups

Zücker Rats aged (16–18) weeks were divided into three groups. The abdomen was cut with a midline incision, and following bile duct cannulation, the portal vein and the splenic and gastroduodenal veins were ligated. After organ recovery, the livers were flushed with IGL-0, IGL-1, or IGL-2 (Table 1) and stored in each solution for 24 h at 4 °C. Animals were randomly distributed in different groups (*n* = 6), as follows:Group 1 (SHAM): Obese Zücker rats underwent transverse laparotomy, and silk ligatures of right suprarenal and diaphragmatic veins and hepatic artery were performed before retrieving the liver.Group 2 (IGL-0 solution): After organ recovery, fatty livers were flushed with 40 mL of IGL-0 preservation solution and were then stored in IGL-0 at 4 °C for 24 h.Group 3 (IGL-1 solution): After organ recovery, fatty livers were flushed with 40 mL of IGL-1 preservation solution and were then stored in IGL-1 at 4 °C for 24 h.Group 4 (IGL-2 solution): After organ recovery, fatty livers were flushed with 40 mL of IGL-1 preservation solution and were then stored in IGL-2 at 4 °C for 24 h.

After 24 h of cold preservation or right after surgery (in the case of Sham), liver samples were rinsed with Ringer’s lactate (20 mL), and samples were taken from the flush. They were then stored at −80 °C for subsequent biochemical determinations.

### 4.3. Biochemical Analyses

#### Transaminase Assay

Liver injury was assessed by alanine aminotransferase (ALT) and aspartate aminotransferase (AST) commercial kits, purchased from RAL (Barcelona, Spain) following the manufacturer’s instructions. Briefly, 100 mol of effluent washout was added to 1 mL of substrate provided by the commercial kit. Transaminase activity was measured at 340 nm using a UV spectrometer.

### 4.4. Glutamate Dehydrogenase (GLDH) Activity

Mitochondrial damage was measured by GLDH activity, following the manufacturer’s instructions of the commercial kit purchased from RANDOX (Crumlin, United Kingdom).

### 4.5. Energy Metabolism (ATP Breakdown)

The determination of ATP in liver samples homogenized in a perchloric acid solution was performed using the ATP assay kit for fluorimetry (Sigma Aldrich ATP colorimetric/fluorometric assay kit, Madrid, Spain). The ATP concentration was determined by the phosphorylation of glycerol, which is a detectable product for the fluorimeter (excitation/emission 535 nm/587 nm) at 37 °C and proportional to the amount of ATP in the sample. Energy breakdown during cold storage was measured through the changes in ATP levels.

### 4.6. 4-Hydroxynonenal Protein Adducts Assay

4-Hydroxynonenal (4-HNE) protein adducts were measured in liver homogenate using the OxiSelect™ HNE Adduct Competitive ELISA Kit (Cell Biolabs, Inc. San Diego, CA, USA). Liver was homogenized in 10% (*w*/*v*) with a Teflon bar in a RIPA solution, (Tris 50 M pH 7.4, 1% Triton 100×, NaCl 150 mM, NaF 5 M, 0.1% sodium dodecyl sulphate, and 1% sodium deoxycholate) with antiprotease solution (aprotinin at 1.7 mg/mL, 2 µg/mL pepstatin, 2 µg/mL leupeptin and 1 mM phenylmethylsulfonyl fluoride, and sodium orthovanadate at 1 mM). The suspension was centrifuged at 2000 g for 5 min and the pellet discarded. Liver homogenates were added to an HNE conjugate preabsorbed ELISA plate. After a brief incubation, an anti-HNE polyclonal antibody was added, followed by an HRP conjugated secondary antibody. The quantity of HNE adduct in protein samples was determined by comparing its absorbance with that of a known HNE-BSA standard curve.

### 4.7. Advanced Oxidation Protein Products (AOPP)

Advanced oxidation protein products (AOPP) are biomarkers of oxidative damage to proteins, detecting dityrosine-containing and cross-linking protein products. The formation of AOPP in the liver homogenates was spectrophotometrically measured at 340 nm. Results were obtained through a standard calibration curve using 100 μL of chloramine-T solution (0–100 μmol/L). AOPP concentration was expressed in nmol/mg protein. Advanced oxidation protein products (AOPPs) in the liver were assayed by a modification of Witko-Sarsat’s method [38].

### 4.8. Glutathione Analysis

Reduced glutathione (GSH) was measured in the liver extracts using the procedure previously described [28]. Liver samples were homogenized in cold buffer containing 5 mm phosphate–EDTA buffer (pH 8.0) and 25% HPO3. The homogenates were ultra-centrifuged at 100,000× *g* and 4 °C for 30 min, and the resulting supernatant, with the fluorescent probe o-phthalaldehyde, was used to determine GSH concentration. Fluorescence was determined at a wavelength emission of 420 nm and excitation at 350 nm. Results are expressed as GSH nmol/mg protein.

### 4.9. Nitrite/Nitrate Analysis

NO production in the liver was determined by tissue accumulation of nitrite and nitrate using a colorimetric assay kit (Cayman, Tallinn, Estonia) according to the manufacturer’s instructions.

### 4.10. Western Blot Analysis

#### ALDH2, Beclin-1, and LC3B

Separated on 6–15% sodium dodecyl sulfate polyacrylamide gel electrophoresis (SDS-PAGE) gels, proteins were blotted into poly-vinylidene fluoride (PVDF) membranes (Bio-Rad, Madrid, Spain) and immunoblotted overnight at 4 °C using antibodies against ALDH2 (Abcam, Cambridge, UK. ref: ab133306), Beclin-1 (Sigma Aldrich, San Louis, Missouri, ref: SAB5700251), and LC3B (Abcam, Cambridge, UK. ref: ab48394). Detection was performed with anti-IgG-HRP (Santa Cruz Biotechnology, Inc., Heidelberg, Germany). In all cases, the chemiluminescence signals were quantified ChemiDoc (Bio-Rad, Madrid, Spain). Both β-actin (Abcam ref: ab8226) and α-tubulin (Abcam ref: ab7291) were used as loading controls.

### 4.11. 4-HNE Protein Adducts and eNOS

Liver samples were homogenized in RIPA (50 M Tris (pH 7.4), 1% Triton X-100, 150 mM NaCl, 5 M NaF, 0.1% sodium dodecyl sulfate, and 1% sodium deoxycholate) and centrifuged for 20 min at 10,000 g. The supernatant was denatured with the addition of Bromophenol Blue (1/2) and heating at 95 °C for 5 min. A total of 50 mg of protein per sample was loaded onto the 10% agarose gel, and wet blotting was carried out on a PVDF membrane (Bio-rad, Irvine, CA, USA). The membranes were blocked for 1 h in Odyssey^®^ Blocking Buffer (LI-COR Biosciences GmbH, Germany) diluted in Tris Base Buffer (TBS, Tris-buffered saline) (pH = 7.4) with 0.05% Tween (TTBS). The membranes were incubated overnight with anti-4-hydroxynonenal (4-HNE) protein adducts and anti-eNOS (BD Biosciences-Europe) antibodies according to the manufacturers’ recommendations.

Detection and analysis were carried out by incubation with secondary fluorescence (800 W) with the Odyssey^®^ Fc system (LI-COR Biosciences GmbH). To quantify the expression, Image Studio 5.2.5 software (LI-COR Biosciences) was used, correcting for the total protein analyzed with the REVERTTM solution (Li-COR Biosciences) according to the manufacturer’s protocol and expressing the results as a percentage with respect to the sham group.

### 4.12. Statistics

Data are expressed as mean ± standard error and were compared statistically by variance analysis, followed by the Student–Newman–Keuls test using GraphPad Prism version 8.1.0 for Windows (GraphPad Prism software, San Diego, CA, USA, 2018) and one-way ANOVA. A level of *p* < 0.05 was considered significant. Significant differences between groups are represented with different letters in the graphs. A group labeled with a letter has significant statistical differences compared to a group labeled with a consecutive letter with a *p* < 0.05.

## Figures and Tables

**Figure 1 ijms-22-05332-f001:**
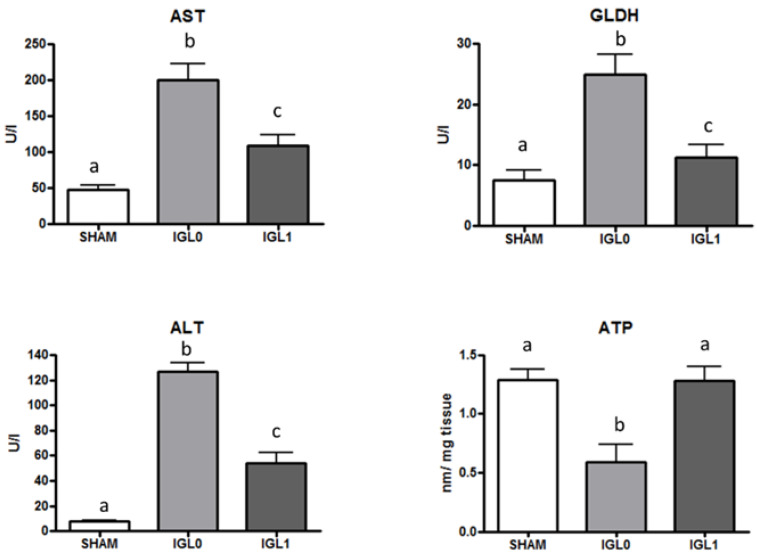
Transaminases (AST/ALT), mitochondrial damage (measured as GLDH), and ATP levels in steatotic livers preserved in IGL-0 (without PEG35) and IGL-1 solutions (PEG35: 1 g/L) vs. SHAM. Results are expressed as mean ± SEM (*n* = 6). Different lowercase letters indicate significant differences among treatments *p* < 0.05.

**Figure 2 ijms-22-05332-f002:**
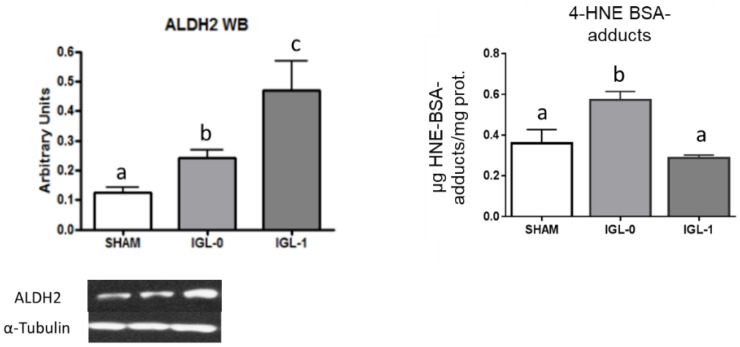
ALDH2 expression and 4-HNE protein-adducts (expressed as µg 4-HNE-BSA-adducts/mg protein) levels in steatotic livers preserved in IGL-0 (no PEG35) and IGL-1 solutions (PEG35: 1 g/L) vs. SHAM. Results are expressed as mean ± SEM (*n* = 6). Different lowercase letters indicate significant differences among treatments *p* < 0.05.

**Figure 3 ijms-22-05332-f003:**
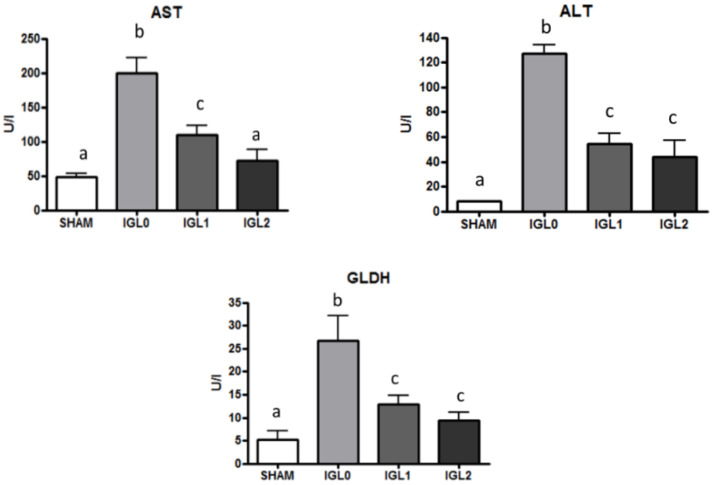
AST, ALT, and GLDH levels in steatotic liver samples for SHAM, IGL-0, IGL-1, and IGL-2 groups. Results are expressed as mean ± SEM (*n* = 6). Different lowercase letters indicate significant differences among treatments *p* < 0.05.

**Figure 4 ijms-22-05332-f004:**
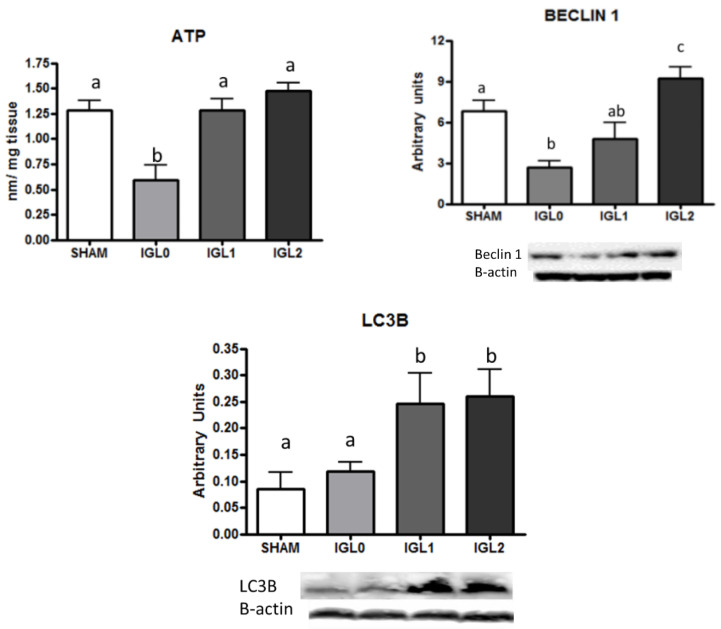
ATP, Beclin-1, and LC3B in steatotic liver samples for SHAM, IGL-0, IGL-1, and IGL-2 groups. Results are expressed as mean ± SEM (*n* = 6). Different lowercase letters indicate significant differences among treatments (one-way ANOVA, *p* < 0.05).

**Figure 5 ijms-22-05332-f005:**
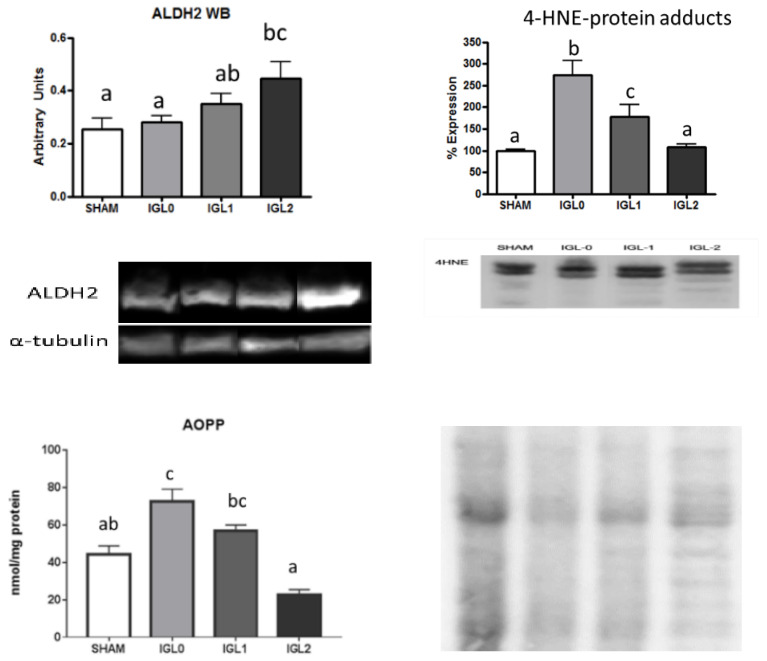
ALDH2 expression and levels of AOPP and 4HNE protein adducts in steatotic liver samples for SHAM, IGL-0, IGL-1, and IGL-2 groups. Results are expressed as mean ± SEM (*n* = 6). Different lowercase letters indicate significant differences among treatments (one-way ANOVA, *p* < 0.05).

**Figure 6 ijms-22-05332-f006:**
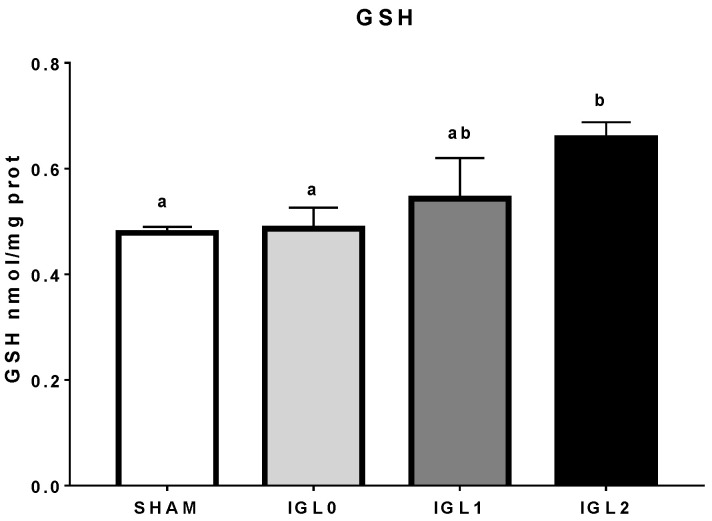
Reduced glutathione (GSH) in IGL-2 solution compared to IGL-1 and IGL-0 (no PEG35) in steatotic liver samples. Different lowercase letters indicate significant differences among treatments (one-way ANOVA, *p* < 0.05).

**Figure 7 ijms-22-05332-f007:**
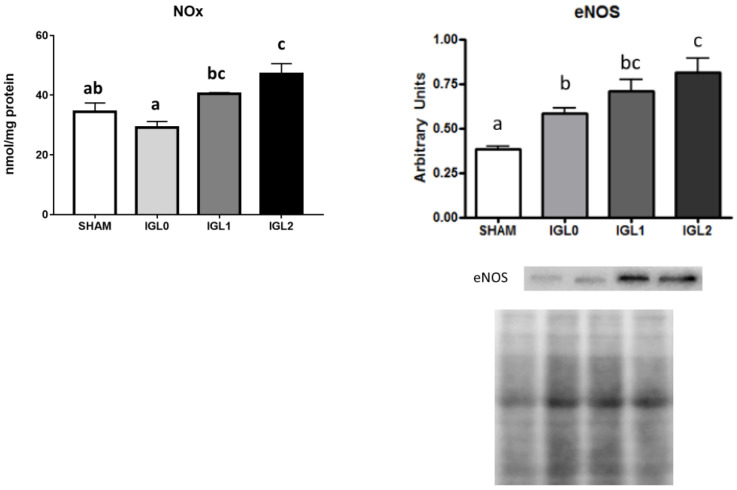
NOx and eNOS in steatotic liver samples for SHAM, IGL-0, IGL-1, and IGL-2 groups. Results are expressed as mean ± SEM (*n* = 6). Different lowercase letters indicate significant differences among treatments (one-way ANOVA, *p* < 0.05).

**Table 1 ijms-22-05332-t001:** Composition of IGL-2 and IGL-1 solutions.

Preservation Solution	IGL-1	IGL-2
**Electrolytes (mmol/L)**		
K^+^	25	25
Na^+^	125	125
Mg^2+^	5.5	
SO_4_		5.5
Zn^2+^		0.091
**Buffers (mmol/L)**		
Phosphate	25	25
Histidine		30
**Impermeants (mmol/L)**		
Mannitol	60	60
Lactobionic acid	80	
**Colloids (g/L)**		
Polyethylene glycol- 35	1	5
**Antioxydants (mmol/L)**		
Glutathione	3	9
**Metabolic precursors (mmol/L)**		
Adenosine	5	5
NaNO_2_ (nmol/L)		50
pH	7.4	7.4
Osmolarity (mosmol/L)	320	320
Viscosity (cP)	1.2	1.4

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
