# Peer review of "Role of PEG35, Mitochondrial ALDH2, and Glutathione in Cold Fatty Liver Graft Preservation: An IGL-2 Approach"

_ijms, 2021, doi:10.3390/ijms22105332_

Round 1
Reviewer 1 Report
Comments to the Author
Thank you for this opportunity to review the manuscript entitled "Role of PEG35, mitochondrial ALDH2 and glutathione in cold fatty liver graft preservation: an IGL-2 approach" submitted by Dr. Gómez-Bardallo et al describing unique and important viewpoint. Surgeons will come across the situation that switching the solutions from one to another near future, when the machine perfusion becomes more casual choice. Although the results showed IGL-2- associated reduction of cold preservation injury evidenced by the decreased liver enzymes leakage in the flush solution, neither prognosis nor peak value of the injury markers (6-12 hours after reperfusion) was shown. Since prediction of post-transplant graft function and injury has become one of the most important areas in the field of machine perfusion related research, many investigators have challenged to find out predictive markers before transplantation. In other words, evaluation of the graft flush solution alone is unsatisfactory. I have the following concerns.
Major issues
This paper lacks some indispensable information. The author should describe and evaluate following issues. I believe that the finding presented here may provide some clues to resolve cold preservation injury and post-operative mortality when the authors analyze more broadly and profoundly.
1) The variance of the steatosis grade is important for the graft quality before and after cold preservation with or without interventions. The author described the graft conditions of Zucker rat (11 w) as “a rate of steatosis of between 60% and 70%” without any microscopic images. The precise methods of the steatosis grading should be presented. It directly connects to the reliability of the whole study.
2) Data duplication (Figures 1, 3, and 4).The author should avoid duplication of data presentation. a) The reviewer cannot understand the difference of ATP data presented as figures 1 and 4. b) The data of liver enzymes (ALT, AST, and GLDH) is presented as a table and figure 3.
3) Autophagy has not evaluated. (Figure 4). The authors described that “As shown in Figure 6, we evidenced a significant upregulation of cytoprotective autophagy in IGL-2 group compared with the others, which correlated with ATP levels in PEG35 groups, showing a positive tendance in ATP prevention for IGL solutions.” The author should not claim anything regarding autophagy and protective autophagy only by the data of Beclin1. Unless assessing other ATGs or autophagolysosome, the flux of autophagy cannot be clarified. The reviewer strongly recommends the evaluations of other ATGs or other chaperons associating autophagy.
4) Precise description of 4-HNE is necessary. The author described that “Figure 2. ALDH2 and 4-HNE protein expression levels in steatotic livers---”. This may be inappropriate. The 4-HNE is a degradation product of phospholipid side chain fatty acid. It is not a protein. The author cannot understand which antibody did the author use, against 4-HNE or 4-HNE-protein adduct (modified protein). Since the data was presented as SDS-PAGE (WB), it would be latter. If so, the caption should be revised as “4-HNE protein adduct” and internal control band should be presented. If it is 4-HNE, the author cannot understand how the band appeared by SDS-PAGE. Precise methods should be described.
5) Graft weight change should be presented. Since the present paper evaluated the efficacy of PEG35 (oncotic agent), graft water content is one of the most important parameters.
Minor issues
Materials and Methods.
6) Animals: Male or female, housing, feeding (chow), fasting etc. should be described.
7) L288-L295: Description of the experimental groups are confusing. The description “Group 2 (IGL-0 solution): Same as Group 2 but fatty livers were flushed with 40 ml of IGL-0 preservation solution” Does not make sense.
8) Did the AOPP assay performed not by a commercially available kit? If so, appropriate reference should be cited. If using a commercial kit, it should be disclosed.
Results
9) The quality of all figures should be improved. Resize the text in the figures. It is hard to read.
10) The author described “a, b, c” to show the statistical significance. But the explanation for the marks (a, b, c) is missing in the materials and methods (statistics section) and figure legends. It should be described properly.
11) WB: Internal control should be presented. Figures 2, 5 : “4-HNE”, Figure 7: eNOS
12) GSH/GSSG: The unit of GSH/GSSG should be described. It may be mol/mol. If so, the value 0.2 (mol/mol) is extremely low value compared to that observed in normal rat liver. Further, the value did not change by cold preservation (SHAM vs. IGL-0, IGL-1). The legitimacy of the data (SHAM and preserved fatty liver) should be discussed.
13) GSH/GSSG: The author described as “There is a significant difference of reduced glutathione in IGL-2 solution, which is expectable due to the initial increased dosage of it in however, high variance in IGL-1 group might suggest that part of its glutathione (if compared to IGL-0) might be spared (therefore, not oxidized) due to other antioxidant capacities derived from PEG35 (See Figure 6).”
The author did not present GSH content but GSH/GSSG ratio. Unless presenting the data of GSH content, the term “reduced glutathione” should not be used. It is confusing.
14) Reference style should be revised according to the journal author guide.
Author Response
Answers to the reviewer 1 (manuscript with changes uploaded)
Comments to the Author
Thank you for this opportunity to review the manuscript entitled "Role of PEG35, mitochondrial ALDH2 and glutathione in cold fatty liver graft preservation: an IGL-2 approach" submitted by Dr. Gómez-Bardallo et al describing unique and important viewpoint. Surgeons will come across the situation that switching the solutions from one to another near future, when the machine perfusion becomes more casual choice. Although the results showed IGL-2- associated reduction of cold preservation injury evidenced by the decreased liver enzymes leakage in the flush solution, neither prognosis nor peak value of the injury markers (6-12 hours after reperfusion) was shown. Since prediction of post-transplant graft function and injury has become one of the most important areas in the field of machine perfusion related research, many investigators have challenged to find out predictive markers before transplantation. In other words, evaluation of the graft flush solution alone is unsatisfactory. I have the following concerns.
This paper lacks some indispensable information. The author should describe and evaluate following issues. I believe that the finding presented here may provide some clues to resolve cold preservation injury and post-operative mortality when the authors analyze more broadly and profoundly.
Comments to the Reviewer: First of all, thank you very much for your time and considerations. Regarding with the question of the prognostic values we agree that this is very important information to explore given the whole context of transplantation. However, we did not provide data regarding the prognostic of the organ because the aim of this study its one step before that, and it is more related with previous studies including hypothermic static preservation, our research topic since 2006. In the direction of the reviewer considerations, we are currently performing experiments with static preservation followed by ex-vivo liver perfusion as a following step in the evaluation of the benefits of IGL-2. We are aware of the importance to predict the outcome of a liver after the reperfusion (especially after machine perfusion), but the present study focusses solely in the static preservation, and the improvement of preservation solutions for this specific step, although in a future, this data can be used for further steps, involving reperfusion, transplantation, etc. For us, the optimization of preservation solutions and their adaptation to static preservation or machine perfusion it is already a challenge, and we rather study that phase before expanding the studies in further consequences.
Major issues
- The variance of the steatosis grade is important for the graft quality before and after cold preservation with or without interventions. The author described the graft conditions of Zucker rat (11 w) as “a rate of steatosis of between 60% and 70%” without any microscopic images. The precise methods of the steatosis grading should be presented. It directly connects to the reliability of the whole study.
We appreciate the comments. When reviewing the protocols, we confirm that by mistake we put 11 weeks age of the rats instead of 16-18 weeks that is what has always been used in the lab. The model used in this study is a reliable and standardized model of moderate steatosis and has been widely used in previous studies published from our group (Please see,
Serafin A. et al. Ischemic Preconditioning Increases the Tolerance of Fatty Liver to Hepatic Ischemia-Reperfusion Injury in the Rat. American Journal of Pathology, Vol. 161, No. 2, August 2002.; Ben Mosbah I et al. Preservation of Steatotic Livers in IGL-1 Solution. Liver transplantation 12:1215-1223, 2006; Panisello-Roselló A et al. Aldehyde Dehydrogenase 2 (ALDH2) in Rat Fatty Liver Cold Ischemia Injury. .Int J Mol Sci. 2018 Aug 22;19(9):2479. doi: 10.3390/ijms19092479) and related . Moreover, in the Serafin et al. reference microscopic evaluation of fat in the graft is provided. This brand of rats provided by Charles River is well characterized and monitored with small deviations, therefore, it is safe to assume that given certain age and weight, a range of fat deposition is guaranteed. Moreover, it is well accepted that 60% of steatosis is a limit for considering a liver non-transplantable.
- Data duplication (Figures 1, 3, and 4). The author should avoid duplication of data presentation. a) The reviewer cannot understand the difference of ATP data presented as figures 1 and 4. b) The data of liver enzymes (ALT, AST, and GLDH) is presented as a table and figure 3.
We understand that providing data in two different formats might be confusing. Sorry for the inconveniences. It was due just an aesthetics matter and compressing the space. Therefore, we proceed to convert all of them to the same format. The reason why part of the data is repeated is to reflect the sequential progress of the study in a coherent manner, since the initial findings lead us to the expansion of the groups of study with IGL2, and without those steps it could be confusing to understand. Thanks for your valuable comments.
- Autophagy has not evaluated.(Figure 4). The authors described that “As shown in Figure 6, we evidenced a significant upregulation of cytoprotective autophagy in IGL-2 group compared with the others, which correlated with ATP levels in PEG35 groups, showing a positive tendance in ATP prevention for IGL solutions.” The author should not claim anything regarding autophagy and protective autophagy only by the data of Beclin1. Unless assessing other ATGs or autophagolysosome, the flux of autophagy cannot be clarified. The reviewer strongly recommends the evaluations of other ATGs or other chaperons associating autophagy.
We welcome the reviewer's suggestion and will consider for our future research. We regret to inform that with the short time provided, it was impossible to obtain ATG. Acquisition of certain material is challenging under the outgoing pandemic situation.
According to reviewer considerations we have measured LC3B as an additional marker of autophagy. The presence of PEG is determinant for activating LC3B although increased concentrations of PEG were not significant.
- Precise description of 4-HNE is necessary. The author described that “Figure 2.ALDH2 and 4-HNE protein expression levels in steatotic livers---”. This may be inappropriate. The 4-HNE is a degradation product of phospholipid side chain fatty acid. It is not a protein. The author cannot understand which antibody did the author use, against 4-HNE or 4-HNE-protein adduct (modified protein). Since the data was presented as SDS-PAGE (WB), it would be latter. If so, the caption should be revised as “4-HNE protein adduct” and internal control band should be presented. If it is 4-HNE, the author cannot understand how the band appeared by SDS-PAGE. Precise methods should be described.
We apologize for giving the misleading information. Indeed, the results refer to 4-HNE-protein adduct. In the first set of experiments, in Figure 2 Hydroxynonenal (HNE) protein-adducts were measured in liver homogenate using the OxiSelect™ HNE Adduct Competitive ELISA Kit (Cell Biolabs, Inc. San Diego, CA). In the Figure 5 we used anti-4-hydroxynonenal (4-HNE) conjugates (Novus Biologicals, Littleton, CO, USA). The text was included in Experimental section.
- Graft weight change should be presented. Since the present paper evaluated the efficacy of PEG35 (oncotic agent), graft water content is one of the most important parameters.
The aim of the present investigations on PEG35 is focused on its incidence in protective cell signaling during cold ischemic insult in hypothermic static conditions. In fact, the target of investigations along the last 15 years in our group has been to explore the activation of cellular protective mechanism exerted by PEG (see PubMed Rosello-Catafau).
The oncotic effect of PEG is well known since many years. In this sense, we agree with the relevance of reviewer comment, especially when other oncotic agents such as HES is present (UW and UW MP) or not (Celsior, HTK) in preservation solutions. Accordingly, your valuable comment, in further studies we will evaluate it, comparing IGL2 (PEG35) vs other solutions such as UW(HES), HTK and Celsior (absence of oncotic agent), respectively. These investigations will be a matter to considered in future SCI publication.
Minor issues
- Animals: Male or female, housing, feeding (chow), fasting etc. should be described.
According to reviewer’s suggestion, data has been included.
- L288-L295: Description of the experimental groups are confusing. The description “Group 2 (IGL-0 solution): Same as Group 2 but fatty livers were flushed with 40 ml of IGL-0 preservation solution”. Does not make sense.
We apologize for that, and it has been amended.
- Did the AOPP assay performed not by a commercially available kit? If so, appropriate reference should be cited. If using a commercial kit, it should be disclosed.
Advanced oxidation protein products (AOPPs) in tissue livers were assayed by a modification of Witko-Sarsat’s method. Specifications have been included in materials and methods.
Results
- The quality of all figures should be improved. Resize the text in the figures. It is hard to read.
According to the reviewer’s suggestion, figures have been resized.
- The author described “a, b, c” to show the statistical significance. But the explanation for the marks (a, b, c) is missing in the materials and methods (statistics section) and figure legends. It should be described properly.
As suggested by the reviewer, the definition of significant differences between groups represented by letters were clarified in materials and methods.
- WB: Internal control should be presented. Figures 2, 5: “4-HNE”, Figure 7: eNOS
As loading controls, we usually include housekeeping proteins, such as α- tubulin or total protein. To quantify the expression of 4-HNE protein-adducts and eNOS, the Image Studio 5.2.5 software (LI-COR Biosciences) was used, and we correct the values for the total protein analyzed with the REVERT (TM) solution (Li-COR Biosciencies) according to the manufacturer's protocol. It has been previously published that total protein stain had many advantages (Kirshner ZZ, Gibbs RB. Use of the REVERT® total protein stain as a loading control demonstrates significant benefits over the use of housekeeping proteins when analyzing brain homogenates by Western blot: An analysis of samples representing different gonadal hormone states. Mol Cell Endocrinol. 2018 Sep 15;473:156-165. doi: 10.1016/j.mce.2018.01.015).
12) GSH/GSSG: The unit of GSH/GSSG should be described. It may be mol/mol. If so, the value 0.2 (mol/mol) is extremely low value compared to that observed in normal rat liver. Further, the value did not change by cold preservation (SHAM vs. IGL-0, IGL-1). The legitimacy of the data (SHAM and preserved fatty liver) should be discussed.
We appreciate your feedback and excellent suggestions on glutathione values. It has allowed us to detect that the figure of ratio was wrong. According to your comments, the figure 6 has been changed to show reduced glutathione measured in liver homogenates and expressed in nmol GSH/ mg of protein. These values are within the limits, although lower than those of non-steatotic livers previously published by us. (Carbonell et al; Oxidative Medicine and Cellular Longevity, vol. 2016, https://doi.org/10.1155/2016/9324692; Alva N, Bardallo RG, et al; Int J Mol Sci. 2018 Mar 29;19(4):1023. doi: 10.3390/ijms19041023.).
We would expect cold-preserved groups to consume glutathione reserves. If there are no differences with respect to sham, it is indicating that adding glutathione to the medium has a positive effect. And in fact, we see more glutathione in the IGL-2 group, which has more than the other two.
13) GSH/GSSG: The author described as “There is a significant difference of reduced glutathione in IGL-2 solution, which is expectable due to the initial increased dosage of it in however, high variance in IGL-1 group might suggest that part of its glutathione (if compared to IGL-0) might be spared (therefore, not oxidized) due to other antioxidant capacities derived from PEG35 (See Figure 6).” The author did not present GSH content but GSH/GSSG ratio. Unless presenting the data of GSH content, the term “reduced glutathione” should not be used. It is confusing.
According to your suggestions and in order to not confuse the reader, we just presented reduced glutathione.
Reviewer 2 Report
The scarcity of organs continues to be a major obstacle to the greater application of liver transplantation. This manuscript reports the protective role of PEG35 in cold storage preservation of marginal organs such as fatty livers. High PEG35 concentration and GSH in IGL-1 increased ALDH, GSH/GSSG and eNOS and reduced 4-HNE and oxidized proteins.
Minor concerns
- The concentrations in PEG35 and GSH should be added in the abstract.
- Table I: too low resolution. Retype the table 1.
- Line 105: insert a sentence with an explanation of the experimental groups considered thus the sentence “The presence of PEG35…” will be clear.
- Line 120: (see Figure 2): modify in (Figure 2).
- Fig 2: The results of 4-HNE assay have not reported in the text as well as the WB bands in the figure, please add.
- Line 131: As it can be seen below (Table 1)……: Table 1 is reported above. Delete (see Table 1) line 133.
- Uniform the units used for 4-HNE quantification (Fig 2 and Fig 5) and add the description of 4-HNE and protein assay in the method section.
- Line 181: (see Figure 6): modify in (Figure 6).
- The resolution of all the figures should be improved as well as the size of the Figures (the description of X and Y axes should be readable).
- In all the legends the use of “steatotic livers” should be reported.
Author Response
Answers to the reviewer 2 (modified manuscript uploaded)
- The concentrations in PEG35 and GSH should be added in the abstract.
According to the reviewer’s suggestion, PEG35 and GSH concentrations has been added in the abstract.
- Table I: too low resolution. Retype the table 1.
Table 2 was done again. We hope that resolution issues are solved this time.
- Line 105: insert a sentence with an explanation of the experimental groups considered thus the sentence “The presence of PEG35…” will be clear.
Thank you for your attention. Changes were accordingly done.
- Line 120: (see Figure 2): modify in (Figure 2).
Considerations were noted and proper amendments were done.
- Fig 2: The results of 4-HNE assay have not reported in the text as well as the WB bands in the figure, please add.
We apologize for this forgetfulness. in Figure 2 Hydroxynonenal (HNE) protein-adducts were measured in liver homogenate using the OxiSelect™ HNE Adduct Competitive ELISA Kit (Cell Biolabs, Inc. San Diego, CA). The figure has been corrected and results are expressed in µg HNE/ mg protein
- Line 131: As it can be seen below (Table): Table 1 is reported above. Delete (see Table 1) line 133.
Thank you for noticing. Mistake has been amended.
- Uniform the units used for 4-HNE quantification (Fig 2 and Fig 5) and add the description of 4-HNE and protein assay in the method section.
We apologize for giving such a misleading information. The first set of experiments, in Figure 2 Hydroxynonenal (HNE) protein-adducts were measured in liver homogenate using the OxiSelect™ HNE Adduct Competitive ELISA Kit (Cell Biolabs, Inc. San Diego, CA). In the Figure 5 we used anti-4-hydroxynonenal (4-HNE) conjugates (Novus Biologicals, Littleton, CO, USA).
In the methods section, information about 4-Hydroxynonenal protein-adducts assay and Western Blotting of 4-HNE protein-adducts and eNOS has been added.
- Line 181: (see Figure 6): modify in (Figure 6).
Changes were done accordingly to the reviewer’s instructions.
- The resolution of all the figures should be improved as well as the size of the Figures (the description of X and Y axes should be readable).
Figures were resized, although we don’t know how to improve the resolution. Our most sincere apologies.
- In all the legends the use of “steatotic livers” should be reported.
As the reviewer suggested, changes were made accordingly.
Round 2
Reviewer 1 Report
I have reviewed the revised manuscript ijms-1179131 entitled "Role of PEG35, mitochondrial ALDH2 and glutathione in cold fatty liver graft preservation: an IGL-2 approach" submitted to IJMS by Dr. Gómez-Bardallo et al.
The authors addressed all issues raised by the reviewer. I fully agree to the idea that basic knowledge and technical advances in the cold storage are important even in the upcoming MP era. Further, it is indeed honorable for me to have an opportunity to involve in an academic activity in the field of organ preservation under such tough days due to COVID19.
Although the revised manuscript is improved, some description and data presentation are still confusing. The quality of the figures should be improved.
I have the following concerns.
Major
Comment 1
I checked 3 papers and found that the authors precisely evaluated the steatosis level in male Zucker rat of 16-18 weeks as the “60-70% steatosis model”. It seemed stable and reliable model. If the experiments were performed using 16-18 weeks old rats in this study, it is the same model. If so, it seems better citing the paper including animal feedings and pathology (Serafin A et al. 2002). In this case, it is a minor issue.
If rats of 11 week-old were used in this study, steatosis grade should be presented, as a supplemental table or figures. Because it is contradictory to the previous papers. In this case, it is a major issue.
Comment 2
Regarding ALDH2, it is still confusing. The authors described the method of ALDH2 activity assay (L425-). I cannot find the data. Please explain the data. It is one of the main data.
Minor
Comment 3
Regarding 4-HNE-protein adducts in Figure 2 (ELISA) and Figure 5 (WB), I understand what the authors did. Using total protein density as the loading control is proper way. Thank you. But, I should point out again that you evaluated 4-HNE-protein adducts by ELISA and WB, not the “4-HNE”. Please describe as 4-HNE-protein adducts instead of 4-HNE; L473 and Figures 2 and 5 (text in the figure).
Comment 4
In addition, the method (L481-L483) is not correct. Please revise as follows if it is what you did.
“The membranes were incubated overnight with anti-4-hydroxynonenal (4-HNE)-protein adducts and anti-eNOS antibodies (BD Biosciences-Europe) according to manufacturers' recommendations.”
Comment 5
Please describe the supplier of all antibodies; ALDH2, Beclin-1, LC3B, beta-actin, and alpha-tubulin.
Comment 6
The authors described the method of water content (L490-). I cannot find the data. Please explain the data.
Comment 7
Statistics
The author added the description. But, I still cannot understand. Please describe the definition of the letters (a,b,c).
Comment 8
L429: Please correct 4-Hidroxynonenal as 4-Hydroxynonenal.
Author Response
Open Review
(x) I would not like to sign my review report
( ) I would like to sign my review report
English language and style
( ) Extensive editing of English language and style required
( ) Moderate English changes required
(x) English language and style are fine/minor spell check required
( ) I don't feel qualified to judge about the English language and style
|
Yes |
Can be improved |
Must be improved |
Not applicable |
|
|
Does the introduction provide sufficient background and include all relevant references? |
(x) |
( ) |
( ) |
( ) |
|
Is the research design appropriate? |
(x) |
( ) |
( ) |
( ) |
|
Are the methods adequately described? |
( ) |
(x) |
( ) |
( ) |
|
Are the results clearly presented? |
( ) |
( ) |
(x) |
( ) |
|
Are the conclusions supported by the results? |
( ) |
(x) |
( ) |
( ) |
Comments and Suggestions for Authors
I have reviewed the revised manuscript ijms-1179131 entitled "Role of PEG35, mitochondrial ALDH2 and glutathione in cold fatty liver graft preservation: an IGL-2 approach" submitted to IJMS by Dr. Gómez-Bardallo et al.
The authors addressed all issues raised by the reviewer. I fully agree to the idea that basic knowledge and technical advances in the cold storage are important even in the upcoming MP era. Further, it is indeed honourable for me to have an opportunity to involve in an academic activity in the field of organ preservation under such tough days due to COVID19.
Before anything else, we would like to honestly thank the reviewer time, commitment and detail. Regardless of the outcome of the decision, we definitely think the manuscript wouldn’t have reached this level of accuracy without their help.
Although the revised manuscript is improved, some description and data presentation are still confusing. The quality of the figures should be improved.
We would like to remark that we published several articles in this same Journal using the same programs and tool to generate the figures and never had problems. We could consult the editor to see if there is anything we can do.
I have the following concerns.
Major
Comment 1
I checked 3 papers and found that the authors precisely evaluated the steatosis level in male Zucker rat of 16-18 weeks as the “60-70% steatosis model”. It seemed stable and reliable model. If the experiments were performed using 16-18 weeks old rats in this study, it is the same model. If so, it seems better citing the paper including animal feedings and pathology (Serafin A et al. 2002). In this case, it is a minor issue.
If rats of 11-week-old were used in this study, steatosis grade should be presented, as a supplemental table or figures. Because it is contradictory to the previous papers. In this case, it is a major issue.
Thanks for your comments. The fact that this is a multigroup study sometimes involves some miscommunication, therefore, the peer review and exhaustive revision is very helpful to polish details such as this. Thanks for noticing it. We had the mistake of changing the sentence in the methods regarding the age of rats. Since we always have used this 16-18 aged rats, as in the case of the present study, we mentioned the references in the answer (Serafin A et al. 2002).
Comment 2
Regarding ALDH2, it is still confusing. The authors described the method of ALDH2 activity assay (L425-). I cannot find the data. Please explain the data. It is one of the main data.
Thanks for your comments but by mistake we didn’t modify the part in the article. We measured levels of ALDH2 protein and described the western blot properly. Consequently, this matter has been amended in the paper text, methods section.
Minor
Comment 3
Regarding 4-HNE-protein adducts in Figure 2 (ELISA) and Figure 5 (WB), I understand what the authors did. Using total protein density as the loading control is proper way. Thank you. But I should point out again that you evaluated 4-HNE-protein adducts by ELISA and WB, not the “4-HNE”. Please describe as 4-HNE-protein adducts instead of 4-HNE; L473 and Figures 2 and 5 (text in the figure).
Comments have been taken into account and changes have been made.
Comment 4
In addition, the method (L481-L483) is not correct. Please revise as follows if it is what you did.
Comments have been taken into account and changes have been made.
Comment 5
Please describe the supplier of all antibodies; ALDH2, Beclin-1, LC3B, beta-actin, and alpha-tubulin.
Supplier of the antibodies have been provided.
Comment 6
The authors described the method of water content (L490-). I cannot find the data. Please explain the data.
Thanks for your comments and sorry for the inconvenience. We realized minutes right after submitting the manuscript about it and we tried to reach the editor regarding this matter. This final version was not received and this was deleted already because of the reasons exposed at this moment. Therefore, we keep this change in the final version we submit now for your consideration.
Comment 7
Statistics
The author added the description. But I still cannot understand. Please describe the definition of the letters (a,b,c).
An explanatory sentence has been added in materials and methods: “A group labeled with a letter has significant statistical differences with a group labeled with and immediate superior or inferior letter of the alphabet with a p < 0.05.” In other words, “different lowercase letters indicate significant differences among experimental groups”, as indicated by the Figure’s footers which have been changed accordingly. An example of this notation style can be found in the following article: Fernandes de Barros Marangoni, L., Ferrier-Pagès, C., Rottier, C. et al. Unravelling the different causes of nitrate and ammonium effects on coral bleaching. Sci Rep 10, 11975 (2020). https://doi.org/10.1038/s41598-020-68916-0.
Comment 8
L429: Please correct 4-Hidroxynonenal as 4-Hydroxynonenal.
Comments have been taken into account and changes have been made.
Submission Date
26 March 2021
Date of this review

Round 3
Reviewer 1 Report
RE: Manuscript: ijms-1179131.R2
I have reviewed the revised manuscript ijms-1179131 entitled "Role of PEG35, mitochondrial ALDH2 and glutathione in cold fatty liver graft preservation: an IGL-2 approach" submitted to IJMS by Dr. Gómez-Bardallo et al.
The authors addressed all issues raised by the reviewer. I think current paper is worthy for publication in the IJMS after minor revision.
Minor
Figure 2 is presented with the title (4-HNE) and the unit (µg HNE/mg protein). Although the author described the subject as “4-HNE protein-adducts in the figure legend, it is still confusing. The standard of the ELISA is not 4-HNE but 4-HNE-BSA adducts.
Comment 1: Figure 2.
Therefore, to avoid misleading, please revise the unit (µg HNE/mg protein) in the figure as follows; (µg 4-HNE-BSA-adducts/mg protein) or (µg 4-HNE-BSA-adducts/mg prot.) or (µg HNE-BSA-adducts/mg protein) or (µg HNE-BSA-adducts/mg prot.) or (µg HNE-BSA/mg protein) or (µg HNE-BSA/mg prot.).
Comment2: Figure legend.
Please include the content “Results are expressed as µg 4-HNE-BSA-adducts/mg protein”.
Author Response
Open Review
(x) I would not like to sign my review report
( ) I would like to sign my review report
English language and style
( ) Extensive editing of English language and style required
( ) Moderate English changes required
( ) English language and style are fine/minor spell check required
(x) I don't feel qualified to judge about the English language and style
Comments and Suggestions for Authors
RE: Manuscript: ijms-1179131.R2
I have reviewed the revised manuscript ijms-1179131 entitled "Role of PEG35, mitochondrial ALDH2 and glutathione in cold fatty liver graft preservation: an IGL-2 approach" submitted to IJMS by Dr. Gómez-Bardallo et al.
The authors addressed all issues raised by the reviewer. I think current paper is worthy for publication in the IJMS after minor revision.
Minor
Figure 2 is presented with the title (4-HNE) and the unit (µg HNE/mg protein). Although the author described the subject as “4-HNE protein-adducts in the figure legend, it is still confusing. The standard of the ELISA is not 4-HNE but 4-HNE-BSA adducts.
Comment 1: Figure 2.
Therefore, to avoid misleading, please revise the unit (µg HNE/mg protein) in the figure as follows; (µg 4-HNE-BSA-adducts/mg protein) or (µg 4-HNE-BSA-adducts/mg prot.) or (µg HNE-BSA-adducts/mg protein) or (µg HNE-BSA-adducts/mg prot.) or (µg HNE-BSA/mg protein) or (µg HNE-BSA/mg prot.).
Comment2: Figure legend.
Please include the content “Results are expressed as µg 4-HNE-BSA-adducts/mg protein”.
All considerations have been taken into account and changes have been made accordingly.
Submission Date
26 March 2021
Date of this review
13 May 2021 05:39:33